# Discrete Contrastive Diffusion for Cross-Modal Music and Image Generation

**Ye Zhu**
Department of Computer Science
Illinois Institute of Technology
Chicago, IL 60616, USA
`yzhu96@hawk.iit.edu`

**Yu Wu**
School of Computer Science
Wuhan University
Wuhan 430000, China
`wuyucs@whu.edu.cn`

**Kyle Olszewski, Jian Ren, Sergey Tulyakov**
Snap Inc.
Santa Monica, CA 90405, USA
`{kolszewski,jren,stulyakov}@snap.com`

**Yan Yan**
Department of Computer Science
Illinois Institute of Technology
Chicago, IL 60616, USA
`yyan34@iit.edu`

## Abstract

Diffusion probabilistic models (DPMs) have become a popular approach to conditional generation, due to their promising results and support for cross-modal synthesis. A key desideratum in conditional synthesis is to achieve high correspondence between the conditioning input and generated output. Most existing methods learn such relationships implicitly, by incorporating the prior into the variational lower bound. In this work, we take a different route—we explicitly enhance input-output connections by maximizing their mutual information. To this end, we introduce a *Conditional Discrete Contrastive Diffusion (CDCD)* loss and design two contrastive diffusion mechanisms to effectively incorporate it into the denoising process, combining the diffusion training and contrastive learning for the first time by connecting it with the conventional variational objectives. We demonstrate the efficacy of our approach in evaluations with diverse multimodal conditional synthesis tasks: dance-to-music generation, text-to-image synthesis, as well as class-conditioned image synthesis. On each, we enhance the input-output correspondence and achieve higher or competitive general synthesis quality. Furthermore, the proposed approach improves the convergence of diffusion models, reducing the number of required diffusion steps by more than *35%* on two benchmarks, significantly increasing the inference speed.

## 1 Introduction

Generative tasks that seek to synthesize data in different modalities, such as audio and images, have attracted much attention. The recently explored diffusion probabilistic models (DPMs) Sohl-Dickstein et al. (2015b) have served as a powerful generative backbone that achieves promising results in both unconditional and conditional generation Kong et al. (2020); Mittal et al. (2021); Lee & Han (2021); Ho et al. (2020); Nichol & Dhariwal (2021); Dhariwal & Nichol (2021); Ho et al. (2022); Hu et al. (2021). Compared to the unconditional case, conditional generation is usually applied in more concrete and practical cross-modality scenarios, *e.g.*, video-based music generation Di et al. (2021); Zhu et al. (2022a); Gan et al. (2020a) and text-based image generation Gu et al. (2022); Ramesh et al. (2021); Li et al. (2019); Ruan et al. (2021). Most existing DPM-based conditional synthesis works Gu et al. (2022); Dhariwal & Nichol (2021) learn the connection between the conditioning and the generated data implicitly by adding a prior to the variational lower bound Sohl-Dickstein et al. (2015b). While such approaches still feature high generation fidelity, the correspondence between the conditioning and the synthesized data can sometimes get lost, as illustrated in the right column in Fig. 1. To this end, we aim to explicitly enhance the input-output faithfulness via their maximized mutual information under the diffusion generative framework for conditional settings in this paper. Examples of our synthesized music audio and image results are given in Fig. 1.

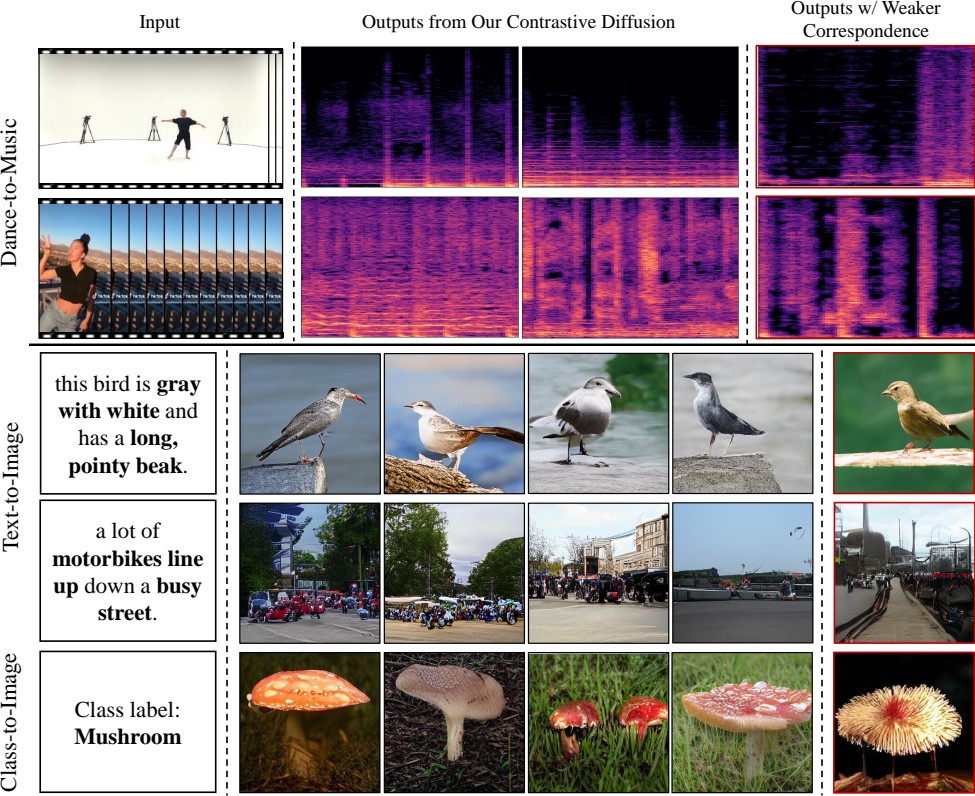

Figure 1: Examples of the input (left column) and synthesized output (middle column) from our contrastive diffusion model for dance-to-music (Rows 1-2), text-to-image (Rows 3-4), and class-conditioned (Row 5) generation experiments on five datasets. The right column shows some synthesized data with reasonable quality but weaker correspondence to the input from existing methods Zhu et al. (2022a); Gu et al. (2022).

Contrastive methods Oord et al. (2018); Bachman et al. (2019); Song & Ermon (2020a) have been proven to be very powerful for data representation learning. Their high-level idea aims to learn the representation $z$ of raw data $x$ based on the assumption that a properly encoded $z$ benefits the ability of a generative model $p$ to reconstruct the raw data given $z$ as prior. This idea can be achieved via optimization of the density ratio $\frac{p(x|z)}{p(x)}$ Oord et al. (2018) as an entirety, without explicitly modeling the actual generative model $p$. While the direct optimization of mutual information via generative models $p$ is a challenging problem to implement and train Song & Ermon (2020b); Belghazi et al. (2018) in the conventional contrastive representation learning field, we show that this can be effectively done within our proposed contrastive diffusion framework. Specifically, we reformulate the optimization problem for the desired conditional generative tasks via DPMs by analogy to the above embedding $z$ and raw data $x$ with our conditioning input and synthesized output. We introduce a *Conditional Discrete Contrastive Diffusion (CDCD)* loss, and design two contrastive diffusion mechanisms - *step-wise parallel diffusion* that invokes multiple parallel diffusion processes during contrastive learning, and *sample-wise auxiliary diffusion*, which maintains one principal diffusion process, to effectively incorporate the *CDCD* loss into the denoising process. We demonstrate that with the proposed contrastive diffusion method, we can not only effectively train so as to maximize the desired mutual information by connecting the *CDCD* loss with the conventional variational objective function, but also to directly optimize the generative network $p$. The optimized *CDCD* loss further encourages faster convergence of a DPM model with fewer diffusion steps. We additionally present our *intra*- and *inter*-negative sampling methods by providing internally disordered and instance-level negative samples, respectively.

To better illustrate the input-output connections, we conduct main experiments on the novel cross-modal dance-to-music generation task Zhu et al. (2022a), which aims to generate music audio based on silent dance videos. Compared to other tasks such as text-to-image synthesis, dance-to-music

generation explicitly evaluates the input-output correspondence in terms of various cross-modal alignment features such as dance-music beats, genre and general quality. However, various generative settings, frameworks, and applications can also benefit from our contrastive diffusion approach, *e.g.*, joint or separate training of conditioning encoders, continuous or discrete conditioning inputs, and diverse input-output modalities as detailed in Sec. 4. Overall, we achieve results superior or comparable to state-of-the-art on three conditional synthesis tasks: dance-to-music (datasets: AIST++ Tsuchida et al. (2019); Li et al. (2021), TikTok Dance-Music Zhu et al. (2022a)), text-to-image (datasets: CUB200 Wah et al. (2011), MSCOCO Lin et al. (2014)) and class-conditioned image synthesis (dataset: ImageNet Russakovsky et al. (2015)). Our experimental findings suggest three key take-away: ① Improving the input-output connections via maximized mutual information is indeed beneficial for their correspondence and the general fidelity of the results (see Fig. 1 and supplement). ② Both our proposed *step-wise parallel diffusion* with *intra*-negative samples and *sample-wise auxiliary diffusion* with *inter*-negative samples show state-of-the-art scores in our evaluations. The former is more beneficial for capturing the intra-sample correlations, *e.g.*, musical rhythms, while the latter improves the instance-level performance, *e.g.*, music genre and image class. ③ With maximized mutual information, our conditional contrastive diffusion *converge in substantially fewer diffusion steps* compared to vanilla DPMs, while maintaining the same or even superior performance (approximately **35%** fewer steps for dance-to-music generation and **40%** fewer for text-to-image synthesis), thus significantly increasing inference speed.

## 2 BACKGROUND

**Diffusion Probabilistic Models.** DPMs Sohl-Dickstein et al. (2015b) are a class of generative models that learn to convert a simple Gaussian distribution into a data distribution. This process consists of a forward *diffusion* process and a reverse *denoising* process, each consisting of a sequence of $T$ steps that act as a Markov chain. During forward diffusion, an input data sample $x_0$ is gradually "corrupted" at each step $t$ by adding Gaussian noise to the output of step $t-1$. The reverse denoising process, seeks to convert the noisy latent variable $x_T$ into the original data sample $x_0$ by removing the noise added during diffusion. The stationary distribution for the final latent variable $x_T$ is typically assumed to be a normal distribution, $p(x_T) = \mathcal{N}(x_T|0, \mathbf{I})$.

An extension of this approach replaces the continuous state with a discrete one Sohl-Dickstein et al. (2015a); Hoogeboom et al. (2021); Austin et al. (2021), in which the latent variables $x_{1:T}$ typically take the form of one-hot vectors with $K$ categories. The diffusion process can then be parameterized using a multinomial categorical transition matrix defined as $q(x_t|x_{t-1}) = \text{Cat}(x_t; p = x_{t-1}Q_t)$, where $[Q_t]_{ij} = q(x_t = j|x_{t-1} = i)$. The reverse process $p_\theta(x_t|x_{t-1})$ can also be factorized as conditionally independent over the discrete sequences Austin et al. (2021).

In both the continuous and discrete state formulations of DPMs Song & Ermon (2020c); Song et al. (2020b); Kingma et al. (2021); Song et al. (2021); Huang et al. (2021); Vahdat et al. (2021), the denoising process $p_\theta$ can be optimized by the KL divergence between $q$ and $p_\theta$ in closed forms Song et al. (2020a); Nichol & Dhariwal (2021); Ho et al. (2020); Hoogeboom et al. (2021); Austin et al. (2021) via the variational bound on the negative log-likelihood:

$$\mathcal{L}_{\text{vb}} = \mathbb{E}_q[\underbrace{D_{\text{KL}}(q(x_T|x_0)||p(x_T))}_{\mathcal{L}_T} + \sum_{t>1} \underbrace{D_{\text{KL}}(q(x_{t-1}|x_t, x_0)||p_\theta(x_{t-1}|x_t))}_{\mathcal{L}_{t-1}} - \underbrace{\log p_\theta(x_0|x_1)}_{\mathcal{L}_0}]. \quad (1)$$

Existing conditional generation works via DPMs Gu et al. (2022); Dhariwal & Nichol (2021) usually learn the implicit relationship between the conditioning $c$ and the synthesized data $x_0$ by directly adding the $c$ as the prior in (1). DPMs with discrete state space provide more controls on the data corruption and denoising compared to its continuous counterpart Austin et al. (2021); Gu et al. (2022) by the flexible designs of transition matrix, which benefits for practical downstream operations such as editing and interactive synthesis Tseng et al. (2020); Cui et al. (2021); Xu et al. (2021). We hence employ contrastive diffusion using a discrete state space in this work.

**Contrastive Representation Learning.** Contrastive learning uses loss functions designed to make neural networks learn to understand and represent the specific similarities and differences between elements in the training data without labels explicitly defining such features, with *positive* and *negative* pairs of data points, respectively. This approach has been successfully applied in learning representations of high-dimensional data Oord et al. (2018); Bachman et al. (2019); He et al. (2020); Song & Ermon (2020a); Chen et al. (2020); Lin et al. (2021). Many such works seek to maximize the

mutual information between the original data $x$ and its learned representation $z$ under the framework of likelihood-free inference Oord et al. (2018); Song & Ermon (2020a); Wu et al. (2021). The above problem can be formulated as maximizing a density ratio $\frac{p(x|z)}{p(x)}$ that preserves the mutual information between the raw data $x$ and learned representation $z$.

To achieve this, existing contrastive methods Oord et al. (2018); Durkan et al. (2020); He et al. (2020); Zhang et al. (2021) typically adopt a neural network to directly model the ratio as an entirety and avoid explicitly considering the actual generative model $p(x|z)$, which has proven to be a more challenging problem Song & Ermon (2020b); Belghazi et al. (2018). In contrast, we show that by formulating the conventional contrastive representation learning problem under the generative setting, the properties of DPMs enable us to directly optimize the model $p$ in this work, which can be interpreted as the optimal version of the density ratio Oord et al. (2018).

**Vector-Quantized Representations for Conditional Generation.** Vector quantization is a classical technique in which a high-dimensional space is represented using a discrete number of vectors. More recently, Vector-Quantized (VQ) deep learning models employ this technique to allow for compact and discrete representations of music and image data Oord et al. (2017); Razavi et al. (2019); Esser et al. (2021b); Dhariwal et al. (2020); Chen et al. (2022). Typically, the VQ-based models use an encoder-codebook-decoder framework, where the "codebook" contains a fixed number of vectors (entries) to represent the original high dimensional raw data. The encoder transforms the input $x$ into feature embedding that are each mapped to the closest corresponding vector in the codebook, while the decoder uses the set of quantized vectors $z$ to reconstruct the input data, producing $x'$ as illustrated in the upper part of Fig. 2.

In this work, we perform conditional diffusion process on the VQ space (*i.e.*, discrete token sequences) as shown in the bottom part of Fig. 2, which largely reduces the dimensionality of the raw data, thus avoiding the expensive raw data decoding and synthesis. As our approach is flexible enough to be employed with various input and output modalities, the exact underlying VQ model we use depends on the target data domain. For music synthesis, we employ a fine-tuned Jukebox Dhariwal et al. (2020) model, while for image generation, we employ VQ-GAN Esser et al. (2021b). See Sec. 4 for further details. We refer to $z$, the latent quantized representation of $x$, as $z_0$ below to distinguish it from the latent representation at prior stages in the denoising process.

## 3 METHOD

Here we outline our approach to cross-modal and conditional generation using our proposed discrete contrastive diffusion approach, which is depicted in Fig. 2. In Sec. 3.1, we formulate our Conditional Discrete Contrastive Diffusion loss in detail, and demonstrate how it helps to maximize the mutual information between the conditioning and generated discrete data representations. Sec. 3.2 defines two specific mechanisms for applying this loss within a diffusion model training framework, *sample-wise* and *step-wise*. In Sec. 3.3, we detail techniques for constructing negative samples designed to improve the overall quality and coherence of the generated sequences.

Given the data pair $(c, x)$, where $c$ is the conditioning information from a given input modality (*e.g.*, videos, text, or a class label), our objective is to generate a data sample $x$ in the target modality (*e.g.*, music audio or images) corresponding to $c$. In the training stage, we first employ and train a VQ-based model to obtain discrete representation $z_0$ of the data $x$ from the target modality. Next, our diffusion process operates on the encoded latent representation $z_0$ of $x$. The denoising process recovers the latent representation $z_0$ given the conditioning $c$ that can be decoded to obtain the reconstruction $x'$. In inference, we generate $z_0$ based on the conditioning $c$, and decode the latent VQ representation $z_0$ back to raw data domain using the decoder from the pre-trained and fixed VQ decoder.

### 3.1 CONDITIONAL DISCRETE CONTRASTIVE DIFFUSION LOSS

We seek to enhance the connection between $c$ and the generated data $z_0$ by maximizing their mutual information, defined as $I(z_0; c) = \sum_{z_0} p_\theta(z_0, c) \log \frac{p_\theta(z_0|c)}{p_\theta(z_0)}$. We introduce a set of negative VQ sequences $Z' = \{z^1, z^2, ..., z^N\}$, encoded from N negative samples $X' = \{x^1, x^2, ..., x^N\}$, and define $f(z_0, c) = \frac{p_\theta(z_0|c)}{p_\theta(z_0)}$. Our proposed Conditional Discrete Contrastive Diffusion (CDCD) loss is:

$$\mathcal{L}_{\text{CDCD}} := -\mathbb{E}\left[\log \frac{f(z_0, c)}{f(z_0, c) + \Sigma_{z^j \in Z'} f(z_0^j, c)}\right]. \tag{2}$$

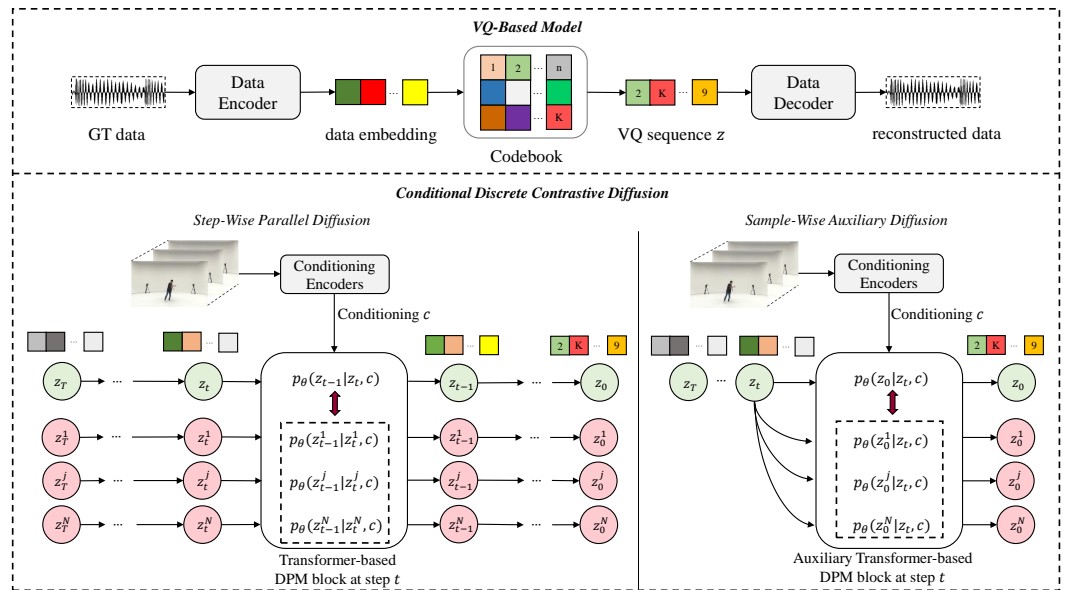

Figure 2: **Overview of the proposed pipeline.** Our framework includes two major components: a VQ-based encoder-decoder model (top) and a conditioned discrete contrastive diffusion as generative model on the VQ space (bottom). In the contrastive diffusion stage, we illustrate our proposed step-wise parallel diffusion (bottom left) and sample-wise auxiliary diffusion (bottom right). The variables in green denote those from the principal diffusion process, while the variables in red represent the diffusion invoked by negative samples. Here we show audio generation from video input, but demonstrate that this approach extends to different modalities, *e.g.*, text-to-image.

The proposed *CDCD* loss is similar to the categorical cross-entropy loss for classifying the positive sample as in Oord et al. (2018), where our conditioning $c$ and the generated data $z_0$ corresponds to the original learned representation and raw data, and optimization of this loss leads to maximization of $I(z_0; c)$. However, the loss in Oord et al. (2018) models the density ratio $f(z_0, c)$ as an entirety. In our case, we demonstrate that the DPMs properties Sohl-Dickstein et al. (2015b); Ho et al. (2020); Austin et al. (2021) enable us to directly optimize the actual distribution $p_\theta$ within the diffusion process for the desired conditional generation tasks. Specifically, we show the connections between the proposed *CDCD* loss and the conventional variational loss $\mathcal{L}_{\mathrm{vb}}$ (see (1)) in Sec. 3.2, and thus how it contributes to efficient DPM learning. Additionally, we can derive the lower bound for the mutual information as $I(z_0; c) \geq \log(N) - \mathcal{L}_{\mathrm{CDCD}}$ (see supplement for details), which indicates that a larger number of negative samples increases the lower bound. These two factors allow for *faster* convergence of a DPM with *fewer* diffusion steps.

## 3.2 PARALLEL AND AUXILIARY DIFFUSION PROCESS

The *CDCD* loss in (2) considers the mutual information between $c$ and $z_0$ in a general way, without specifying the intermediate diffusion steps. We propose and analyze two contrastive diffusion mechanisms to efficiently incorporate this loss into DPM learning, and demonstrate that we can directly optimize the generative model $p_\theta$ in the diffusion process. We present our *step-wise parallel diffusion* and the *sample-wise auxiliary diffusion* mechanisms, which are distinguished by the specific operations applied for the intermediate negative latent variables $z_{1:T}^j$ for each negative sample $x^j$. The high-level intuition for the parallel and auxiliary designs is to emphasize different attributes of the synthesized data given specific applications. Particularly, we propose the parallel variant to learn the internal coherence of the audio sequential data by emphasizing the gradual change at each time step, while the auxiliary mechanism focuses more on the sample-level connections to the conditioning.

**Step-Wise Parallel Diffusion.** This mechanism not only focuses on the mutual information between $c$ and $z_0$, but also takes the intermediate negative latent variables $z_{1:T}^j$ into account by explicitly invoking the complete diffusion process for each negative sample $z^j \in Z'$. As illustrated in Fig. 2 (bottom left), we initiate $N + 1$ parallel diffusion processes, among which $N$ are invoked by negative

samples. For each negative sample $x^j \in X'$, we explicitly compute its negative latent discrete variables $z_{0:T}^j$. In this case, (2) is as follows (see supplement for the detailed derivation):

$$\mathcal{L}_{\text{CDCD-Step}} := \mathbb{E}_Z \log \left[ 1 + \frac{p_\theta(z_{0:T})}{p_\theta(z_{0:T}|c)} N\mathbb{E}_{Z'} \left[ \frac{p_\theta(z_{0:T}^j|c)}{p_\theta(z_{0:T}^j)} \right] \right] \equiv \mathcal{L}_{\text{vb}}(z, c) - C \sum_{z^j \in Z'} \mathcal{L}_{\text{vb}}(z^j, c). \quad (3)$$

The equation above factorizes the proposed *CDCD* loss using the step-wise parallel diffusion mechanism into two terms, where the first term corresponds to the original variational bound $\mathcal{L}_{\text{vb}}$, and the second term can be interpreted as the negative sum of variational bounds induced by the negative samples and the provided conditioning $c$. $C$ is a constant as detailed in our supplement.

**Sample-Wise Auxiliary Diffusion.** Alternatively, our *sample-wise auxiliary diffusion* mechanism maintains one principal diffusion process, as in traditional diffusion training, shown in Fig. 2 (bottom right). It contrasts the intermediate *positive* latent variables $z_{1:T}$ with the negative sample $z_0^j \in Z$. In this case, we can write the *CDCD* loss from. (2) as (see supplement for details):

$$\mathcal{L}_{\text{CDCD-Sample}} := \mathbb{E}_q[-\log p_\theta(z_0|z_t, c)] - C \, \Sigma_{z^j \in Z'} \mathbb{E}_q[-\log p_\theta(z_0^j|z_t, c)]. \quad (4)$$

As with the step-wise loss, the *CDCD-Sample* loss includes two terms. The first refers to sampling directly from the positive $z_0$ at an arbitrary timestep $t$. The second sums the same auxiliary loss from negative samples $z_0^j$. This marginalization operation is based on the property of Markov chain as in previous discrete DPMs Austin et al. (2021); Gu et al. (2022), which imposes direct supervision from the sample data. The first term is similar to the auxiliary denoising objective in Austin et al. (2021); Gu et al. (2022).

Both contrastive diffusion mechanisms enable us to effectively incorporate the *CDCD* loss into our DPM learning process by directly optimizing the actual denoising generative network $p_\theta$.

**Final Loss Function.** The final loss function for our contrastive diffusion training process is:

$$\mathcal{L} = \mathcal{L}_{\text{vb}}(z, c) + \lambda \mathcal{L}_{\text{CDCD}}, \quad (5)$$

$\mathcal{L}_{\text{vb}}$ is conditioning $c$ related, and takes the form of $\mathcal{L}_{t-1} = D_{\text{KL}}(q(z_{t-1}|z_t, z_0)||p_\theta(z_{t-1}|z_t, c))$ as in Gu et al. (2022), where $c$ included as the prior for all the intermediate steps. $\mathcal{L}_{\text{CDCD}}$ refers to either the step-wise parallel diffusion or sample-wise auxiliary diffusion loss. Empirically, we can omit the first term in (3), or directly optimize $\mathcal{L}_{\text{CDCD-Step}}$, in which the standard $\mathcal{L}_{\text{vb}}$ is already included. The detailed training algorithm is explained in the supplement.

## 3.3 Intra- and Inter-Negative Sampling

Previous contrastive works construct negative samples using techniques such as image augmentation Chen et al. (2020); He et al. (2020) or spatially adjacent image patches Oord et al. (2018). In this work, we categorize our sampling methods into *intra-* and *inter*-negative samplings as in Fig. 3. For the *intra*-sample negative sampling, we construct $X'$ based on the given original $x$. This bears resemblance to the patch-based technique in the image domain Oord et al. (2018). As for the audio data, we first divide the original audio waveform into multiple chunks, and randomly shuffle their ordering. For the *inter*-sample negative sampling, $X'$ consists of instance-level negative samples $x'$ that differ from the given data pair $(c, x)$. In practice, we define negative samples $x'$ to be music sequences with different musical genres from $x$ in the music generation task, while $x'$ denotes images other than $x$ in the image synthesis task (in practice, we choose $x'$ with different class labels as $x$).

Based on our proposed contrastive diffusion modes and negative sampling methods, there are four possible contrastive settings: step-wise parallel diffusion with either intra- or inter-negative sampling (denoted as *Step-Intra* and *Step-Inter*), or sample-wise auxiliary diffusion with either intra- or inter-negative sampling (denoted as *Sample-Intra* and *Sample-Inter*). Intuitively, we argue that *Step-Intra* and *Sample-Inter* settings are more reasonable compared to *Step-Inter* and *Sample-Intra* because of the consistency between the diffusion data corruption process and the way to construct negative samples. Specifically, the data corruption process in the discrete DPMs includes sampling and replacing certain tokens with some random or mask tokens at each diffusion step Austin et al. (2021); Gu et al. (2022), which is a chunk-level operation within a given data sequence similar to the ways we construct *intra*-negative samples by shuffling the chunk-level orders. In contrast, the *sample-wise auxiliary diffusion* seeks to provide sample-level supervision, which is consistent with our inter-negative sampling method.

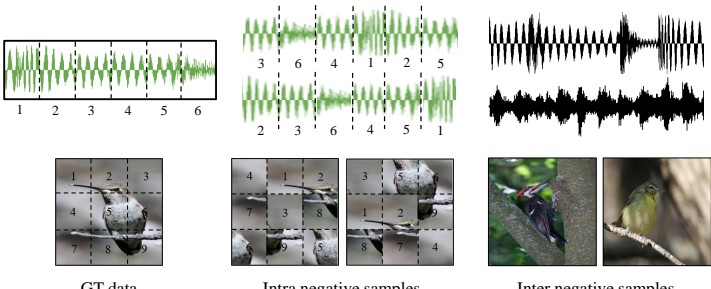

GT data        Intra negative samples        Inter negative samples

Figure 3: Illustration of *intra-* and *inter-*negative sampling for music and image data.

In the interest of clarity and concision, we only present the experimental results for *Step-Intra* and *Sample-Inter* settings in Sec. 4 of our main paper. The complete results obtained with other contrastive settings and more detailed analysis are included in the supplement.

## 4 EXPERIMENTS

We conduct experiments on three conditional generation tasks: dance-to-music generation, text-to-image synthesis, and class-conditioned image synthesis. For the dance-to-music task, we seek to generate audio waveforms for complex music from human motion and dance video frames. For the text-to-image task, the objective is to generate images from given textual descriptions. Given our emphasis on the input-output faithfulness for cross-modal generations, the main analysis are based on the dance-to-music generation task since the evaluation protocol from Zhu et al. (2022a) explicitly measures such connections in terms of beats, genre and general correspondence for generated music.

### 4.1 DANCE-TO-MUSIC GENERATION

**Dataset.** We use the AIST++ Li et al. (2021) dataset and the TikTok Dance-Music dataset Zhu et al. (2022a) for the dance-to-music experiments. AIST++ is a subset of the AIST dataset Tsuchida et al. (2019), which contains 1020 dance videos and 60 songs performed by professional dancers and filmed in clean studio environment settings without occlusions. AIST++ provide human motion data in the form of SMPL Loper et al. (2015) parameters and body keypoints, and includes the annotations for different genres and choreography styles. The TikTok Dance-Music dataset includes 445 dance videos collected from the social media platform. The 2D skeleton data extracted with OpenPose Cao et al. (2017); Cao et al. (2019) is used as the motion representation. We adopt the official cross-modality splits without overlapping music songs for both datasets.

**Implementations.** The sampling rate for all audio signals is 22.5 kHz in our experiments. We use 2-second music samples as in Zhu et al. (2022a) for the main experiments. We fine-tuned the pre-trained Jukebox Dhariwal et al. (2020) for our Music VQ-VAE model. For the motion encoder, we deploy a backbone stacked with convolutional layers and residual blocks. For the visual encoder, we extract I3D features Carreira & Zisserman (2017) using a model pre-trained on Kinectics Kay et al. (2017) as the visual conditioning. The motion and visual encoder outputs are concatenated to form the final continuous conditioning input to our contrastive diffusion model. For the contrastive diffusion model, we adopt a transformer-based backbone to learn the denoising network $p_\theta$. It includes 19 transformer blocks, with each block consisting of full attention, cross attention and feed forward modules, and a channel size of 1024 for each block. We set the initial weight for the contrastive loss as $\lambda = 5e - 5$. The number $N$ of intra- and inter-negative samples for each GT music sample is 10. The visual encoder, motion encoder, and the contrastive diffusion model are jointly optimized. More implementation details are provided in the supplement.

**Evaluations.** The evaluation of synthesized music measures both the conditioning-output correspondence and the general synthesis quality using the metrics introduced in Zhu et al. (2022a). Specifically, the metrics include the beats coverage score, the beats hit scores, the genre accuracy score, and two subjective evaluation tests with Mean Opinion Scores (MOS) for the musical coherence and general quality. Among these metrics, the beats scores emphasize the intra-sample properties, since they calculate the second-level audio onset strength within musical chunks Ellis (2007), while the genre accuracy focuses on the instance-level musical attributes of music styles. Detailed explanations of the above metrics can be found in Zhu et al. (2022a). We compare against multiple dance-to-music

Table 1: Quantitative evaluation results for the dance-to-music task on the AIST++ dataset. This table shows the best performance scores we obtain for different contrastive diffusion steps. We report the mean and standard deviations of our contrastive diffusion for three inference tests.

| Musical features | Rhythms | Rhythms | Genre | Coherence | Quality |
|---|---|---|---|---|---|
| Metrics | Coverage ↑ | Hit ↑ | Accuracy ↑ | MOS ↑ | MOS ↑ |
| GT Music | 100 | 100 | 88.5 | 4.7 | 4.8 |
| Foley | 74.1 | 69.4 | 8.1 | 2.9 | - |
| Dance2Music | 83.5 | 82.4 | 7.0 | 3.0 | - |
| CMT | 85.5 | 83.5 | 11.6 | 3.0 | - |
| D2M-GAN | 88.2 | 84.7 | 24.4 | 3.3 | 3.4 |
| Ours Vanilla | 89.0±1.1 | 83.8±1.5 | 25.3±0.8 | 3.3 | **3.6** |
| Ours Step-Intra | **93.9**±1.2 | **90.7**±1.5 | 25.8±0.6 | **3.6** | 3.5 |
| Ours Sample-Inter | 91.8±1.6 | 86.9±1.4 | **27.2**±0.5 | **3.6** | **3.6** |

Table 2: Quantitative evaluuastion results for the dance-to-music task on the TikTok dataset. We set the default number of diffusion steps to be 80.

| Methods | Beats Coverage/Hit ↑ |
|---|---|
| D2M-GAN | 88.4/ 82.3 |
| Ours Vanilla | 88.7/ 81.4 |
| Ours Step-Intra | **91.8/ 86.3** |
| Ours Sample-Inter | 90.1/ 85.5 |

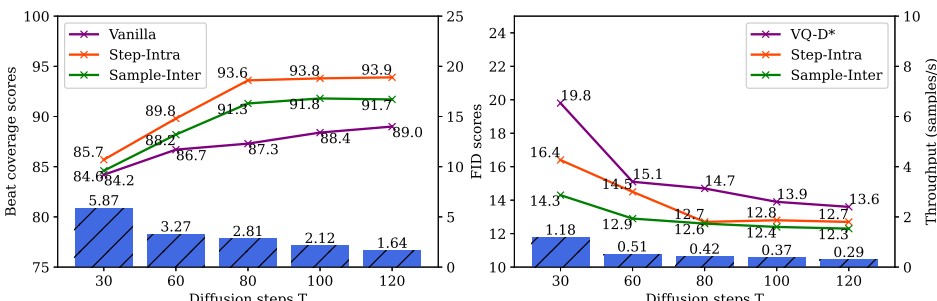

Figure 4: Convergence analysis in terms of diffusion steps for the dance-to-music task on AIST++ dataset (left) and the text-to-image task on CUB200 dataset (right). We observe that our contrastive diffusion models converge at around 80 steps and 60 steps, resulting 35% steps and 40% steps less compared to the vanilla models that converge at 120 steps and 100 steps, while maintaining superior performance, respectively. We use the same number of steps for training and inference.

generation works: Foley Gan et al. (2020a), Dance2Music Aggarwal & Parikh (2021), CMT Di et al. (2021), and D2M-GAN Zhu et al. (2022a). The first three models rely on symbolic discrete MIDI musical representations, while the last one also uses a VQ musical representation. The major difference between the symbolic MIDI and discrete VQ musical representations lies within the fact that the MIDI is pre-defined for each instrument, while the VQ is learning-based. The latter thus enables complex and free music synthesis appropriate for scenarios like dance videos.

**Results and Discussion.** The quantitative experimental results are shown in Tab. 1 and Tab. 2. Our proposed methods achieve better performance than the competing methods even with vanilla version without contrastive mechanisms. Furthermore, we find that the *Step-Intra* setting is more helpful in increasing the beats scores, while the *Sample-Inter* setting yields more improvements for the genre accuracy scores. We believe this is due to the evaluation methods of different metrics. The beats scores measure the chunk-level (*i.e.*, , the audio onset strength Ellis (2007)) consistency between the GT and synthesized music samples Zhu et al. (2022a), while the genre scores consider the overall musical attributes of each sample sequence in instance level. This finding is consistent with our assumptions in Sec. 3.3.

**Convergence Analysis.** We also analyze the impact of the proposed contrastive diffusion on model convergence in terms of diffusion steps. The number of diffusion steps is a significant hyper-parameter for DPMs Sohl-Dickstein et al. (2015b); Nichol & Dhariwal (2021); Austin et al. (2021); Gu et al. (2022); Kingma et al. (2021) that directly influences the inference time and synthesis quality. Previous works have shown that a larger number of diffusion steps usually lead to better model performance, but longer inference times Kingma et al. (2021); Gu et al. (2022). We demonstrate that, with the improved mutual information via the proposed contrastive diffusion method, we can greatly reduce the number of steps needed. As shown in Fig. 4 (left), we observe that the beats scores reach a stable level at approximately 80 steps, ∼35% less than the vanilla DPM that converges in ∼120 steps. More ablation studies and analysis on this task can be found in the supplement.

Table 3: FID and CLIPScore for text-to-image synthesis on CUB-200 and MSCOCO datasets. The VQ-D model with ⋆ shows the results we reproduced by training using the original code, which can be considered as our baseline. We show the results obtained with default 80 diffusion steps in both training and inference.

| Datasets | CUB-200 | | MSCOCO | |
|---|---|---|---|---|
| Metrics | FID ↓ | CLIPScore ↑ | FID ↓ | CLIPScore ↑ |
| StackGAN | 51.89 | - | 74.05 | - |
| StackGAN++ | 15.30 | - | 81.59 | - |
| SEGAN | 18.17 | - | 32.28 | - |
| AttnGAN | 23.98 | - | 35.49 | 65.66 |
| DM-GAN | 16.09 | - | 32.64 | 65.45 |
| DF-GAN | 14.81 | - | 21.42 | 66.42 |
| DAE-GAN | 15.19 | - | 28.12 | - |
| DALLE | 56.10 | 74.66 | 27.50 | - |
| VQ-D (T=100) | 12.97 | - | 30.17 | - |
| VQ-D⋆ (T=80) | 14.61 | 74.96 | 36.45 | 65.53 |
| Ours Step-Intra (T=80) | 12.73 | 75.32 | 32.25 | 66.22 |
| Ours Sample-Inter (T=80) | **12.61** | **75.50** | 28.76 | **66.79** |

Table 4: FID scores and top 5 classification accuracy using pre-trained ResNet101 He et al. (2016) for class-conditioned image synthesis on ImageNet $256 \times 256$. Our model follows the *Sample-Inter* contrastive setting with 80 diffusion steps in both training and inference.

| Methods | FID ↓ | Acc.↑ |
|---|---|---|
| ImageBART | 21.19 | - |
| VQGAN | 15.78 | - |
| IDDPM | 12.30 | - |
| VQ-D (T=100) | 11.89 | 72.72 |
| Ours (T=80) | **11.74** | **77.63** |

## 4.2 Conditional Image Synthesis

**Dataset.** We conduct text-to-image synthesis on CUB200 Wah et al. (2011) and MSCOCO datasets Lin et al. (2014). The CUB200 dataset contains images of 200 bird species. Each image has 10 corresponding text descriptions. The MSCOCO dataset contains 82k images for training and 40k images for testing. Each image has 5 text descriptions. We also perform the class-conditioned image generation on ImageNet Deng et al. (2009); Russakovsky et al. (2015). Implementation details for both tasks are provided in the supplement.

**Evaluations.** We adopt two evaluation metrics for text-to-image synthesis: the classic FID score Heusel et al. (2017) as the general measurement for image quality, and the CLIPScore Hessel et al. (2021) to evaluate the correspondence between the given textual caption and the synthesized image. For the class-conditioned image synthesis, we use the FID score and a classifier-based accuracy for general and input-output correspondence measurement. We compare against text-to-image generation methods including StackGAN Zhang et al. (2017), StackGAN++ Zhang et al. (2018), SEGAN Tan et al. (2019), AttnGAN Xu et al. (2018), DM-GAN Zhu et al. (2019), DF-GAN Tao et al. (2020), DAE-GAN Ruan et al. (2021), DALLE Ramesh et al. (2021), and VQ-Diffusion Gu et al. (2022). For experiments on ImageNet, we list the result comparisons with ImageBART Esser et al. (2021a), VQGAN Esser et al. (2021b), IDDPM Nichol & Dhariwal (2021), and VQ-D Gu et al. (2022). Specifically, VQ-Diffusion Gu et al. (2022) also adopts the discrete diffusion generative backbone, which can be considered as the vanilla version without contrastive mechanisms. Additionally, we provide more comparisons with other methods in terms of dataset, model scale and training time in the supplement for a more comprehensive and fair understanding of our proposed method.

**Results and Discussion.** The quantitative results are represented in Tab. 3 and Tab. 4. We observe that our contrastive diffusion achieves state-of-the-art performance for both general synthesis fidelity and input-output correspondence, and the *Sample-Inter* contrastive setting is more beneficial compared to *Step-Intra* for the image synthesis. This empirical finding again validates our assumption regarding the contrastive settings in Sec. 3.3, where the *Sample-Inter* setting helps more with the instance-level synthesis quality. Notably, as shown in Fig. 4 (right), our contrastive diffusion method shows model convergence at about 60 diffusion steps, while the vanilla version converges at approximately 100 steps on CUB200 Wah et al. (2011), which greatly increases the inference speed by 40%.

## 5 Conclusion

While DPMs have demonstrated remarkable potential, improving their training and inference efficiency while maintaining flexible and accurate results for conditional generation is an ongoing challenge, particularly for cross-modal tasks. Our Conditional Discrete Contrastive Diffusion (*CDCD*) loss addresses this by maximizing the mutual information between the conditioning input and the generated output. Our contrastive diffusion mechanisms and negative sampling methods effectively incorporate this loss into DPM training. Extensive experiments on various cross-modal conditional generation tasks demonstrate the efficacy of our approach in bridging drastically differing domains.

## ACKNOWLEDGMENT

This research is partially supported by NSF SCH-2123521 and Snap unrestricted gift funding. This article solely reflects the opinions and conclusions of its authors and not the funding agents.

## ETHICS STATEMENTS

As in other media generation works, there are possible malicious uses of such media to be addressed by oversight organizations and regulatory agencies. Our primary objective as researchers is always creating more reliable and secure AI and machine learning systems that maximally benefit our society.

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

## A    MORE RELATED WORKS

In addition to the fields of *Diffusion Probabilistic Models*, *Contrastive Representation Learning*, and *VQ Representations for Conditional Generation* discussed in the main paper, our work is also closely related to the multi-modal learning and generation fields.

The research topic of multimodal learning, which incorporates data from various modalities such as audio, vision, and language has attracted much attention in recent years Baltrušaitis et al. (2018); Zhu et al. (2022b); Wu et al. (2023). General audio and visual learning works typically seek to investigate their correlations from the intrinsic synchronization nature Aytar et al. (2016); Korbar et al. (2018); Owens & Efros (2018); Owens et al. (2016); Arandjelovic & Zisserman (2017), and then utilize them in various downstream audio-visual tasks such as audio-visual action recognition Kazakos et al. (2019); Gao et al. (2020), audio-visual event localization and parsing Tian et al. (2018); Zhu et al. (2021a); Wu et al. (2019); Wu & Yang (2021), and audio-visual captioning Rahman et al. (2019); Wang et al. (2018). Works to generate music from visual or/and motion data have also been widely explored in recent years Gan et al. (2020a); Di et al. (2021); Aggarwal & Parikh (2021); Zhu et al. (2022a). For vision and language area, the text generation from visions are extensively explored in the image and video captioning task Zhu et al. (2020; 2021b); Anderson et al. (2018); You et al. (2016); Wang et al. (2017). At the same time, works on image/video generation from text have also attracted much attention with recently released largescale models Radford et al. (2021); Li et al. (2019); Ruan et al. (2021); Ramesh et al. (2021).

## B    DETAILED PROOF AND TRAINING

### B.1    LOWER BOUND OF CDCD LOSS

We show that the proposed *CDCD* loss has a lower bound related to the mutual information and the number of negative samples $N$. The derivations below are similar to those from Oord et al. (2018):

$$\mathcal{L}_{\text{CDCD}} := \mathbb{E}_Z\left[-\log \frac{\frac{p_\theta(z_0|c)}{p_\theta(z_0)}}{\frac{p_\theta(z_0|c)}{p_\theta(z_0)} + \sum_{z^j \in Z'} \frac{p_\theta(z_0^j|c)}{p_\theta(z_0^j)}}\right] \tag{6a}$$

$$= \mathbb{E}_Z \log\left[1 + \frac{p_\theta(z_0)}{p_\theta(z_0|c)} \sum_{z^j \in Z'} \frac{p_\theta(z_0^j|c)}{p_\theta(z_0^j)}\right] \tag{6b}$$

$$\approx \mathbb{E}_Z \log\left[1 + N\frac{p_\theta(z_0)}{p_\theta(z_0|c)} \mathbb{E}_{Z'}\left[\frac{p_\theta(z_0^j|c)}{p_\theta(z_0^j)}\right]\right] \tag{6c}$$

$$= \mathbb{E}_Z \log\left[1 + N\frac{p_\theta(z_0)}{p_\theta(z_0|c)}\right] \tag{6d}$$

$$\geq \mathbb{E}_Z \log\left[N\frac{p_\theta(z_0)}{p_\theta(z_0|c)}\right] \tag{6e}$$

$$= \log(N) - I(z_0, c). \tag{6f}$$

## B.2 CONVENTIONAL VARIATIONAL LOSS

The conventional variational loss $\mathcal{L}_{\mathrm{vb}}$ is derived as follows Sohl-Dickstein et al. (2015b):

$$
\begin{aligned}
\mathcal{L}_{\mathrm{vb}}(x) &:= \mathbb{E}_q[-\log \frac{p_\theta(x_{0:T})}{q(x_{1:T}|x_0)}] \\
&= \mathbb{E}_q[-\log p(x_T) - \sum_{t>1} \log \frac{p_\theta(x_{t-1}|x_t)}{q(x_t|x_{t-1})} - \log \frac{p_\theta(x_0|x_1)}{q(x_1|x_0)}] \\
&= \mathbb{E}_q[-\log p(x_T) - \sum_{t>1} \log \frac{p_\theta(x_{t-1}|x_t)}{q(x_{t-1}|x_t, x_0)} \cdot \frac{q(x_{t-1}|x_0)}{q(x_t|x_0)} - \log \frac{p_\theta(x_0|x_1)}{q(x_1|x_0)}] \\
&= \mathbb{E}_q[-\log \frac{p(x_T)}{q(x_T|x_0)} - \sum_{t>1} \log \frac{p_\theta(x_{t-1}|x_t)}{q(x_{t-1}|x_t, x_0)} - \log p_\theta(x_0|x_1)] \\
&= \mathbb{E}_q[D_{\mathrm{KL}}(q(x_T|x_0)||p(x_T)) + \sum_{t>1} D_{\mathrm{KL}}(q(x_{t-1}|x_t, x_0)||p_\theta(x_{t-1}|x_t)) - \log p_\theta(x_0|x_1)].
\end{aligned}
$$
(7)

## B.3 $\mathcal{L}_{\mathrm{vb}}$ WITH CONDITIONING PRIOR

Following the unconditional conventional variational loss, we then show its conditional variant with the conditioning $c$ as prior, which has also been adopted in Gu et al. (2022).

$$
\begin{aligned}
\mathcal{L}_{\mathrm{vb}}(x, c) &= \mathcal{L}_0 + \mathcal{L}_1 + ... + \mathcal{L}_{T-1} + \mathcal{L}_T \\
\mathcal{L}_0 &= -\log p_\theta(x_0|x_1, c) \\
\mathcal{L}_{t-1} &= D_{\mathrm{KL}}(q(x_{t-1}|x_t, x_0)||p_\theta(x_{t-1}|x_t, c)) \\
\mathcal{L}_T &= D_{\mathrm{KL}}(q(x_T|x_0)||p(x_T))
\end{aligned}
$$
(8)

## B.4 STEP-WISE AND SAMPLE-WISE CONTRASTIVE DIFFUSION

Below, we show the full derivation for the step-wise parallel contrastive diffusion loss. Given that the intermediate variables from $z_{1:T}$ are also taken into account in this step-wise contrastive diffusion, we slightly modify the initial notation $f(z_0, c) = \frac{p_\theta(z_0|c)}{p_\theta(z_0)}$ from Eq.(2) in the main paper to $f(z, c) = \frac{p_\theta(z_{0:T}|c)}{p_\theta(z_{0:T})}$.

$$
\mathcal{L}_{\mathrm{CDCD-Step}} := -\mathbb{E}_Z \left[\log \frac{f(z, c)}{f(z, c) + \sum_{z^j \in Z'} f(z^j, c)}\right]
\tag{9a}
$$

$$
= \mathbb{E}_Z \log \left[1 + \frac{\sum_{z^j \in Z'} f(z^j, c)}{f(z, c)}\right]
\tag{9b}
$$

$$
= \mathbb{E}_Z \log \left[1 + \frac{p_\theta(z_{0:T})}{p_\theta(z_{0:T}|c)} \sum_{z^j \in Z'} \frac{p_\theta(z^j_{0:T}|c)}{p_\theta(z^j_{0:T})}\right]
\tag{9c}
$$

$$
\approx \mathbb{E}_Z \log \left[1 + \frac{p_\theta(z_{0:T})}{p_\theta(z_{0:T}|c)} N\mathbb{E}_{Z'} \frac{p_\theta(z^j_{0:T}|c)}{p_\theta(z^j_{0:T})}\right] \ (same \ as \ Eq.(6c))
\tag{9d}
$$

$$
\approx \mathbb{E}_Z \mathbb{E}_q \log[\frac{q(z_{1:T}|z_0)}{p_\theta(z_{0:T}|c)} N \frac{p_\theta(z_{0:T}|c)}{q(z_{1:T}|z_0)}] \ (conditional \ p_\theta)
\tag{9e}
$$

$$
\approx \mathbb{E}_q[-\log \frac{p_\theta(z_{0:T}|c)}{q(z_{1:T}|z_0)}] - \log N \ \mathbb{E}_{Z'} \mathbb{E}_q[-\log \frac{p_\theta(z_{0:T}|c)}{q(z_{1:T}|z_0)}]
\tag{9f}
$$

$$
= \mathcal{L}_{\mathrm{vb}}(z, c) - C \sum_{z^j \in Z'} \mathcal{L}_{\mathrm{vb}}(z^j, c).
\tag{9g}
$$

---
**Algorithm 1** Conditional Discrete Contrastive Diffusion Training. The referenced equations can be found in the main paper.

---
**Input:** Initial network parameters $\theta$, contrastive loss weight $\lambda$, learning rate $\eta$, number of negative samples $N$, total diffusion steps $T$, conditioning information $c$, contrastive mode $m \in \{Step, \ Sample\}$.

1: **for** each training iteration **do**
2:      $t \sim Uniform(\{1, 2, ..., T\})$
3:      $z_t \leftarrow Sample \ from \ q(z_t|z_{t-1})$
4:      $\mathcal{L}_{\text{vb}} \leftarrow \sum_{i=1,...,t} \mathcal{L}_i$   ▷ Eq. 1
5:      **if** m == *Step* **then**
6:          **for** j = 1, ..., N **do**
7:              $z_t^j \leftarrow Sample \ from \ q(z_t^j|z_{t-1}^j, c)$ ▷ from negative variables in previous steps
8:          **end for**
9:          $\mathcal{L}_{\text{CDCD}} = -\frac{1}{N} \sum \mathcal{L}_{\text{vb}}^j$   ▷ Eq. 3
10:     **else if** m == *Sample* **then**
11:        **for** j = 1, ..., N **do**
12:            $z_t \leftarrow Sample \ from \ q(z_t|z_0^j, c)$ ▷ from negative variables in step 0
13:        **end for**
14:        $\mathcal{L}_{\text{CDCD}} = -\frac{1}{N} \sum \mathcal{L}_{z_0}^j$   ▷ Eq. 4
15:     **end if**
16:     $\mathcal{L} \leftarrow \mathcal{L}_{\text{vb}} + \lambda \mathcal{L}_{\text{CDCD}}$   ▷ Eq. 5
17:     $\theta \leftarrow \theta - \eta \nabla_\theta \mathcal{L}$
18: **end for**

---

In the above Eq.(9g), $C$ stands for a constant that equals to $\log N$, which can be further adjusted by the weight we select for the *CDCD* loss as in Eq. 5.

Similarly for the sample-wise auxiliary contrastive diffusion, the loss can be derived as follows:

$$\mathcal{L}_{\text{CDCD}-\text{Sample}} := -\mathbb{E}_Z \left[\log \frac{f(z_0, c)}{f(z_0, c) + \sum_{z^j \in Z'} f(z_0^j, c)}\right] \tag{10a}$$

$$= \mathbb{E}_Z \log \left[1 + \frac{p_\theta(z_0)}{p_\theta(z_0|c)} \ N \mathbb{E}_{Z'}\left[\frac{p_\theta(z_0^j|c)}{p_\theta(z_0^j)}\right]\right] \tag{10b}$$

$$\approx \mathbb{E}_Z \mathbb{E}_q \log\left[\frac{q(z_{1:T}|z_0)}{p_\theta(z_0|c)} \ N \ \frac{p_\theta(z_0|c)}{q(z_{1:T}|z_0)}\right] \tag{10c}$$

$$\approx \mathbb{E}_q\left[-\log \frac{p_\theta(z_0|c)}{q(z_{1:T}|z_0)}\right] - N \ \mathbb{E}_{Z'}\mathbb{E}_q\left[-\log \frac{p_\theta(z_0|c)}{q(z_{1:T}|z_0)}\right] \tag{10d}$$

$$= \mathbb{E}_q\left[-\log p_\theta(z_0|z_t, c)\right] - C \sum_{z^j \in Z'} \mathbb{E}_q\left[-\log p_\theta(z_0^j|z_t, c)\right]. \tag{10e}$$

Note that from a high-level perspective, our contrastive idea covers two different concepts, while conventional contrastive learning usually focuses only on the negative samples. In our case, due to the unique formulation of diffusion models that bring the diffusion steps into the methodology design, we consider the contrast within the context of "negative samples" and "negative steps" (also corresponds to the "negative intermediate steps"). In the deviation above, we use the symbols $Z$ and $q$ to distinguish between these two concepts.

## B.5   CONDITIONAL DISCRETE CONTRASTIVE DIFFUSION TRAINING

The training process for the proposed contrastive diffusion is explained in Algo. 1.

## C   ADDITIONAL EXPERIMENTAL DETAILS AND ANALYSIS

### C.1   DANCE-TO-MUSIC TASK

**Implementation.** The sampling rate for all audio signals is 22.5 kHz in our experiments. We use 2-second music samples as in Zhu et al. (2022a) for our main experiments, resulting in 44,100 audio

data points for each raw music sequence. For the Music VQ-VAE, we fine-tuned Jukebox Dhariwal et al. (2020) on our data to leverage its pre-learned codebook from a large-scale music dataset (approximately 1.2 million songs). The codebook size $K$ is 2048, with a token dimension $d_z = 128$, and the hop-length $L$ is 128 in our default experimental setting. For the motion module, we deploy a backbone stacked with convolutional layers and residual blocks. The dimension size of the embedding we use for music conditioning is 1024. For the visual module, we extract I3D features Carreira & Zisserman (2017) using a model pre-trained on Kinectics Kay et al. (2017) as the visual conditioning information, with a dimension size of 2048. In the implementation of our contrastive diffusion model, we adopt a transformer-based backbone to learn the denoising network $p_\theta$. It includes 19 transformer blocks, in which each block is consists of full-attention, cross-attention and a feed-forward network, and the channel size for each block is 1024. We set the initial weight for the contrastive loss as $\lambda = 5e - 5$. The numbers of intra- and inter-negative samples for each GT music sample are both 10. The AdamW Loshchilov & Hutter (2017) optimizer with $\beta_1 = 0.9$ and $\beta_2 = 0.96$ is deployed in our training, with a learning rate of $4.5e - 4$. We also employ an adaptive weight for the denoising loss weight by gradually decreasing the weight as the diffusion step increases and approaches the end of the chain. The visual module, motion module, and the contrastive diffusion model are jointly optimized. The architecture of adopted motion encoder is shown in Tab. 5, which is the same as in Zhu et al. (2022a).

Table 5: Architecture for the motion encoder.

| $6 \times 1$, stride=1, Conv 256, LeakyReLU |
| --- |
| Residual Stack 256 |
| $3 \times 1$, stride=1, Conv 512, LeakyReLU |
| Residual Stack 512 |
| $3 \times 1$, stride=1, Conv 1024, LeakyReLU |
| Residual Stack 1024 |
| $3 \times 1$ , stride=1, Conv 1024, LeakyReLU |
| $4 \times 1$, stride=1, Conv 1 |

Other than the aforementioned implementation details, we also include the mask token technique that bears resemblance to those used in language modelling Devlin et al. (2018) and text-to-image synthesis Gu et al. (2022) for our dance-to-music generation task. We adopt a truncation rate of 0.86 in our inference.

**MOS Evaluation Test.** We asked a total of 32 participants to participate in our subjective Mean Opinion Scores (MOS) music evaluations Zhu et al. (2022a); Kumar et al. (2019), among which 11 of them are female, while the rest are male. For the dance-music coherence test, we fuse the generated music samples with the GT videos as post-processing. We then asked each evaluator to rate 20 generated videos with a score of 1 (least coherent) to 5 (most coherent) after watching the processed video clip. Specifically, the participants are asked to pay more attention to the dance-music coherence in terms of the dance moves corresponding to the music genre and rhythm, rather than the overall music quality, with reference to the GT video clips with the original music. As for the overall quality evaluations, we only play the audio tracks without the video frames to each evaluator. As before, they are asked to rate the overall music quality with a score of 1 (worst audio quality) to 5 (best audio quality).

**Training Cost.** For the dance2music task experiments on the AIST++ dataset, we use 4 NVIDIA RTX A5000 GPUs, and train the model for approximately 2 days. For the same task on the TikTok dance-music dataset, the training takes approximately 1.5 days on the same hardware.

**Complete Results for Contrastive Settings.** As discussed in our main paper, there are four possible combinations for contrastive settings given different contrastive diffusion mechanisms and negative sampling methods. Here, we include complete quantitative scores for different contrastive settings in Tab. 6. We observe that all the four contrastive settings, including the *Step-Inter* and *Sample-Intra* settings that are not reported in our main paper, help to improve the performance. As we noted, amongst all the settings, *Step-Intra* and *Sample-Inter* are more reasonable and yield larger improvements for intra-sample data attributes (*i.e.*, beats scores) and instance-level features (*i.e.*, genre accuracy scores).

Table 6: Complete quantitative evaluation results for the dance-to-music generation task on the AIST++ dataset. We report the mean and standard deviations of our contrastive diffusion for three inference tests.

| Musical features | Rhythms | Rhythms | Genre | Coherence | Quality |
|---|---|---|---|---|---|
| Metrics | Coverage ↑ | Hit ↑ | Accuracy ↑ | MOS ↑ | MOS ↑ |
| GT Music | 100 | 100 | 88.5 | 4.7 | 4.8 |
| Foley Gan et al. (2020a) | 74.1 | 69.4 | 8.1 | 2.9 | - |
| Dance2Music Aggarwal & Parikh (2021) | 83.5 | 82.4 | 7.0 | 3.0 | - |
| CMT Di et al. (2021) | 85.5 | 83.5 | 11.6 | 3.0 | - |
| D2M-GAN Zhu et al. (2022a) | 88.2 | 84.7 | 24.4 | 3.3 | 3.4 |
| Ours Vanilla | 89.0±1.1 | 83.8±1.5 | 25.3±0.8 | 3.3 | **3.6** |
| Ours Step-Intra | **93.9**±1.2 | **90.7**±1.5 | 25.8±0.6 | **3.6** | 3.5 |
| Ours Step-Inter | 92.4±1.0 | 88.9±1.7 | 24.3±0.7 | 3.4 | 3.5 |
| Ours Sample-Intra | 91.5±1.7 | 84.6±1.6 | 26.0±0.8 | 3.5 | **3.6** |
| Ours Sample-Inter | 91.8±1.6 | 86.9±1.4 | **27.2**±0.5 | **3.6** | **3.6** |

Table 7: Ablation results for different music lengths on the AIST++ dataset.

| Length | Methods | Beats Coverage ↑ | Beats Hit ↑ | Genre Acc. ↑ |
|---|---|---|---|---|
| 2s | D2M-GAN Zhu et al. (2022a) | 88.2 | 84.7 | 24.4 |
| | Ours Vanilla | 89.0 | 83.8 | 25.3 |
| | Ours Step-Intra | 93.9 | 90.7 | 25.8 |
| | Ours Sample-Inter | 91.8 | 86.9 | 27.2 |
| 4s | D2M-GAN Zhu et al. (2022a) | 87.1 | 83.0 | 23.3 |
| | Ours Vanilla | 86.2 | 81.8 | 24.6 |
| | Ours Step-Intra | 93.1 | 86.4 | 25.3 |
| | Ours Sample-Inter | 91.4 | 83.9 | 26.1 |
| 6s | D2M-GAN Zhu et al. (2022a) | - | - | - |
| | Ours Vanilla | 84.8 | 81.1 | 22.7 |
| | Ours Step-Intra | 87.9 | 83.2 | 22.9 |
| | Ours Sample-Inter | 86.3 | 81.6 | 23.0 |

**Ablation on Music Length.** Although we use 2-second musical sequences in the main experiments to make for consistent and fair comparisons with Zhu et al. (2022a), our framework can also synthesize longer musical sequences. In the supplementary, we show our generated music sequences in 6-seconds. The quantitative evaluations in terms of different musical sequence lengths are presented Tab. 7, where we show better performance when synthesizing longer musical sequences.

## C.2 TEXT-TO-IMAGE TASK

**Implementation.** For the text-to-image generation task, we adopt VQ-GAN Esser et al. (2021b) as the discrete encoder and decoder. The codebook size $K$ is 2886, with a token dimension $d_z = 256$. VQ-GAN converts a $256 \times 256$ resolution image to $32 \times 32$ discrete tokens. For the textual conditioning, we employ the pre-trained CLIP Radford et al. (2021) model to encode the given textual descriptions. The denoising diffusion model $p_\theta$ has 18 transformer blocks and a channel size of 192, which is a similar model scale to the small version of VQ-Diffusion Gu et al. (2022). We use $\lambda = 5e - 5$ as the contrastive loss weight. Similar to the dance-to-music task, we also use the adaptive weight that changes within the diffusion stages. We keep the same truncation rate of 0.86 as in our dance-to-music experiment and in Gu et al. (2022). Unlike in the dance-to-music experiments, where we jointly learn the conditioning encoders, both the VQ-GAN and CLIP models are fixed during the contrastive diffusion training.

**Training Cost.** For the text2image task experiments on the CUB200 dataste, the training takes approximately 5 days using 4 NVIDIA RTX A5000 GPUs. For the same experiments on the MSCOCO dataset, we run the experiments on Amazon Web Services (AWS) using 8 NVIDIA Tesla V100 GPUs. This task required 10 days of training.

Table 8: Comparisons in terms of the task, datasets, model scale, and training time between our proposed CDCD and some other diffusion-based works.

| Tasks | Dataset | Method | Param. | Training time |
|---|---|---|---|---|
| Dance-to-Music | AIST++ Li et al. (2021) | Ours CDCD | 350M | 8 V100 days |
| Dance-to-Music | TikTok Zhu et al. (2022a) | Ours CDCD | 350M | 6 V100 days |
| Text-to-Image | CUB200 Wah et al. (2011) | Ours CDCD | 35M | 20 V100 days |
| Text-to-Image | MSCOCO Lin et al. (2014) | Ours CDCD | 35M | 80 V100 days |
| Class-conditioned | ImageNet | Ours CDCD | 370 M | 160 V100 days |
| Image Synthesis | ImageNet | ADMDhariwal & Nichol (2021) | - | 150-1000 V100 days |
| Image Synthesis | ImageNet | LDM Rombach et al. (2022) | 400M | 270 V100 days |

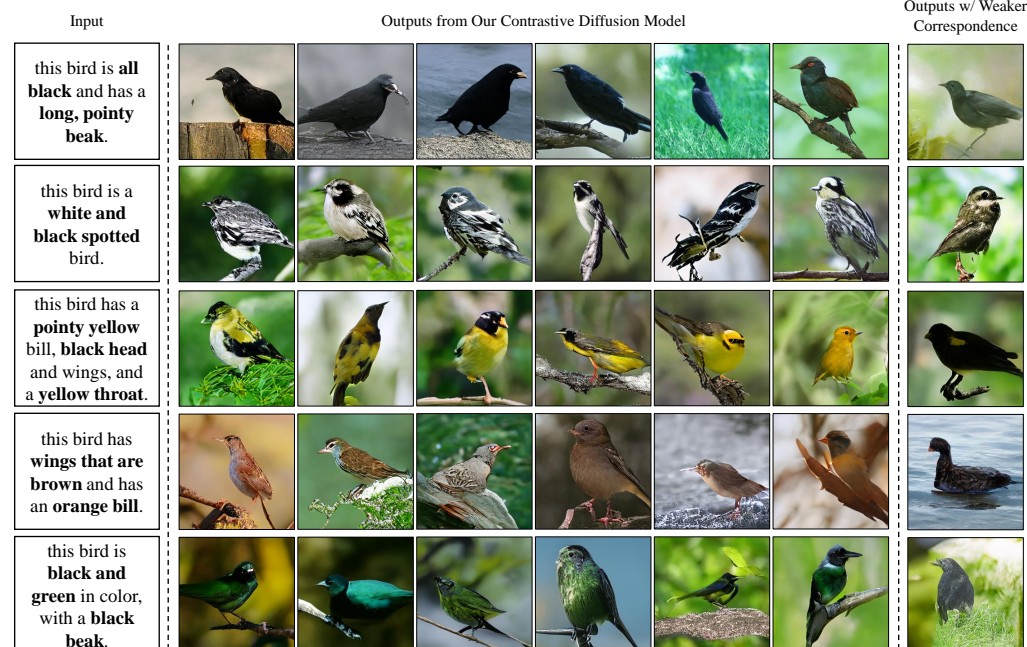

Figure 5: More qualitative results from our text-to-image experiments on CUB200 dataset. We show examples of the text input (left column), the synthesized images from our contrastive diffusion model with 80 diffusion steps and a FID score of 12.61 (middle column), and the output from existing method Gu et al. (2022) with 100 diffusion steps and a FID score 12.97.

## C.3 CLASS-CONDITIONED IMAGE SYNTHESIS TASK

**Implementation.** For the class-conditioned image synthesis, we also adopt the pre-trained VQ-GAN Esser et al. (2021b) as the discrete encoder and decoder. We replace the conditioning encoder with class embedding optimized during the contrastive diffusion training. The size of the conditional embedding is 512. Other parameters and techniques remain the same, as in the text-to-image task.

**Training Cost.** For the class-conditioned experiments on the ImageNet, we use 8 NVIDIA Tesla V100 GPUs running on AWS. This task required 20 days of training.

## D MORE QUALITATIVE RESULTS

### D.1 GENERATED MUSIC SAMPLES

For qualitative samples of synthesized dance music sequences, please refer to our anonymous page in the supplement with music samples. In addition to the generated music samples on AIST++ Tsuchida et al. (2019); Li et al. (2021) and TikTok Dance-Music Dataset Zhu et al. (2022a), we also include some qualitative samples obtained with the music editing operations based on the dance-music genre

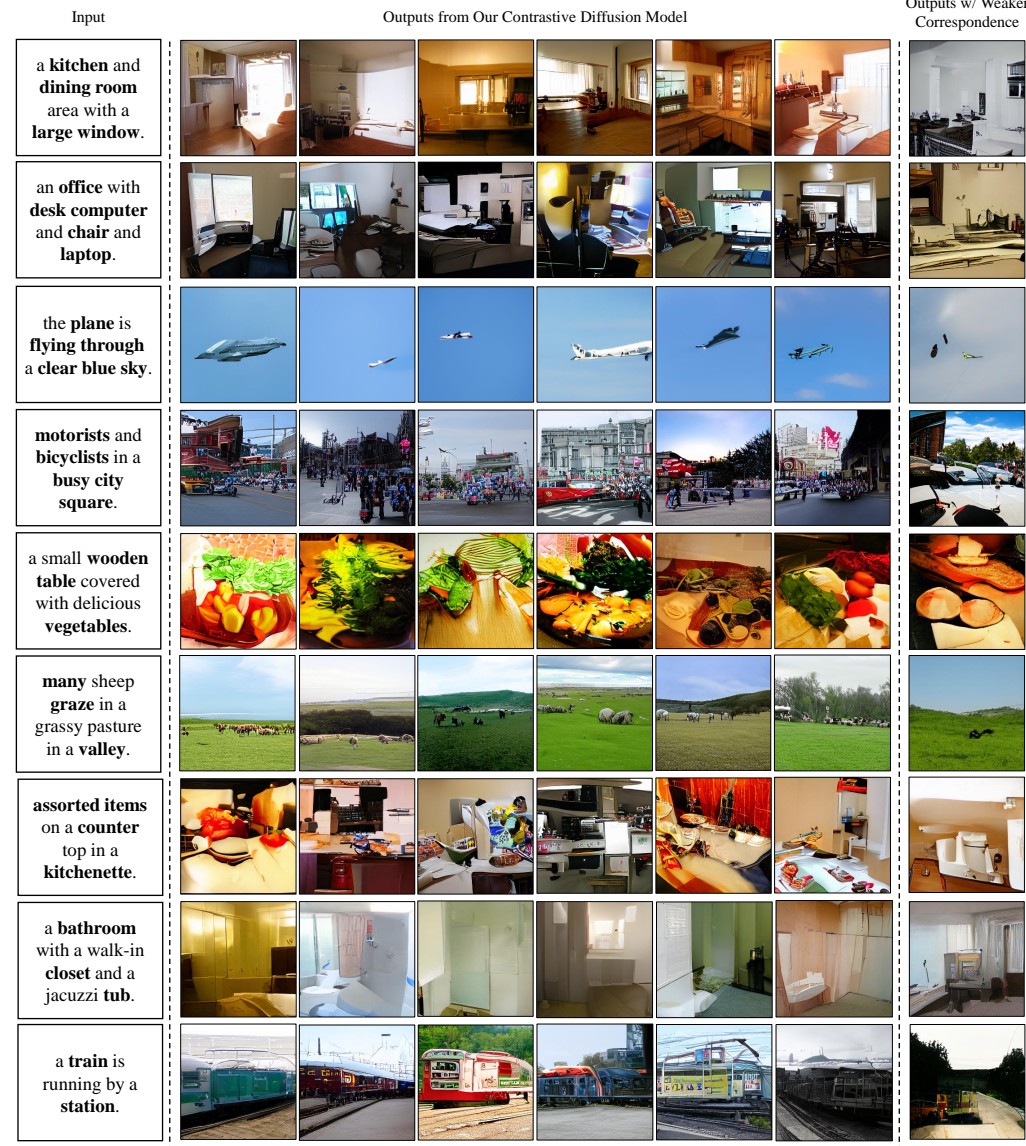

Figure 6: More qualitative results from our text-to-image experiments on COCO dataset. We show examples of the text input (left column), the synthesized images from our contrastive diffusion model with 80 diffusion steps and a FID score of 28.76 (middle column), and the output from existing method Gu et al. (2022) with 100 diffusion steps and a FID score 30.17.

annotations from AIST++. Specifically, we edit the original paired motion conditioning input with a different dance-music genre using a different dance choreographer.

**Discussion on Musical Representations and Audio Quality.** It is worth noting that we only compare the overall audio quality with that of D2M-GAN Zhu et al. (2022a). This is due to the nature of the different musical representations in the literature of deep-learning based music generation Gan et al. (2020a); Dong et al. (2018); Huang et al. (2019); Gan et al. (2020b); Aggarwal & Parikh (2021). There are mainly two categories for adopted musical representations in previous works: pre-defined symbolic and learning-based representations Ji et al. (2020); Briot et al. (2020). For the former symbolic music representation, typical options include 1D piano-roll and 2D MIDI-based representations. While these works benefit from the pre-defined music synthesizers and produce music that does not include raw audio noise, the main limitation is that such representations are

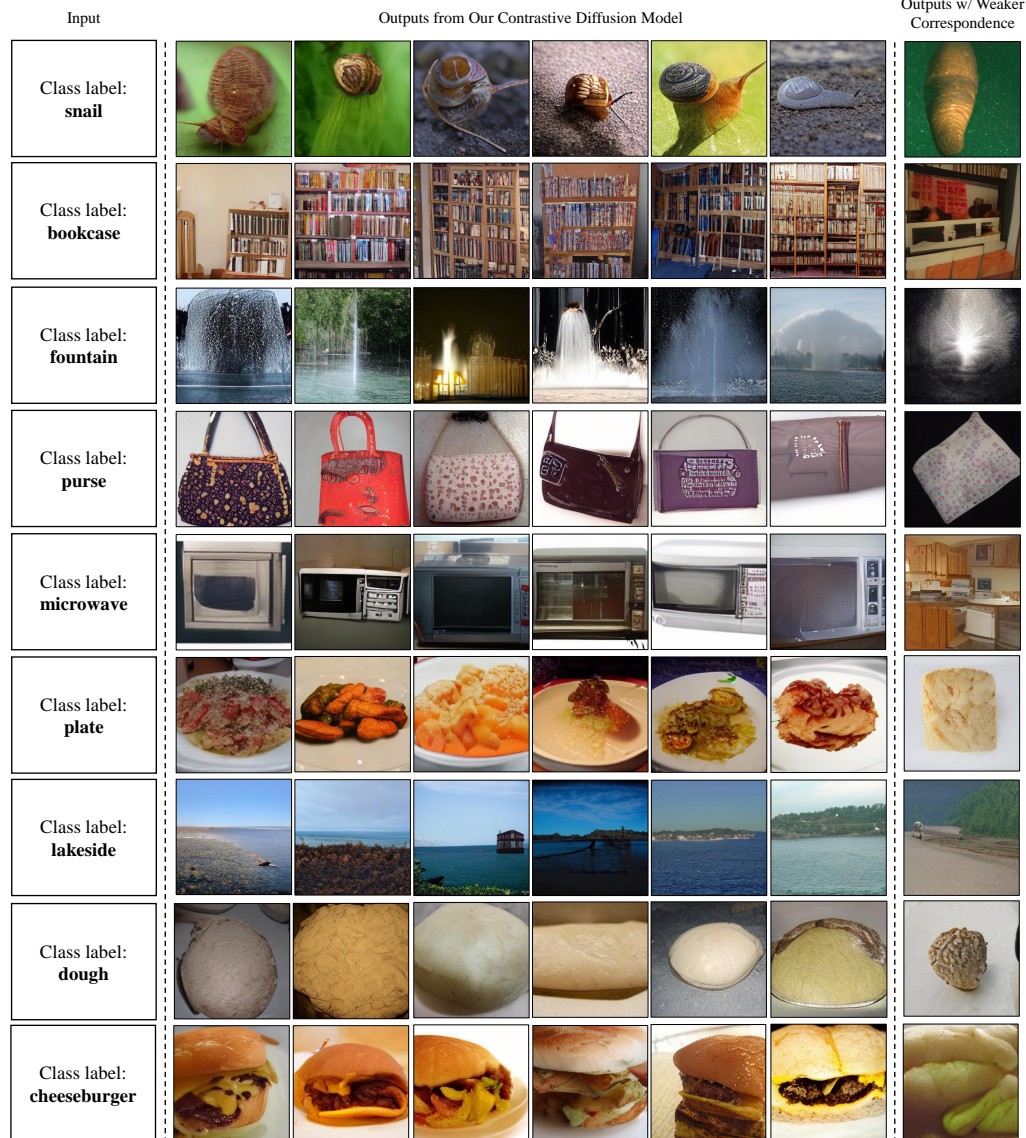

Figure 7: More qualitative results from our class-conditioned image synthesis experiments on ImageNet. We show examples of the text input (left column), the synthesized images from our contrastive diffusion model with 80 diffusion steps and a FID score of 11.74 (middle column), and the output from existing method Gu et al. (2022) with 100 diffusion steps and a FID score 11.89.

usually limited to a single specific instrument, which hinders their flexibility to be applied in wider and more complex scenarios such as dance videos. In contrast, the learning-based music representations (*i.e.*, musical VQ in our case) rely on well-trained music synthesizers as decoders, but can be used as a unified representation for various musical sounds, *e.g.*, instruments or voices. However, the training of such music encoders and decoders for high-quality audio signals itself remains a challenging problem. Specifically, high-quality audio is a form of high-dimensional data with an extremely large sampling rate, even compared to high-resolution images. For example, the sampling rate for CD-quality audio signals is 44.1 kHz, resulting in 2,646,000 data points for a one-minute musical piece. To this end, existing deep learning based works Dhariwal et al. (2020); Kumar et al. (2019) for music generation employ methods to reduce the number of dimensions, *e.g.*, by introducing hop

lengths and a smaller sampling rate. These operations help to make music learning and generation more computationally tractable, but also introduce additional noise in the synthesized audio signals.

In this work, we adopt the pre-trained JukeBox model Dhariwal et al. (2020) as our music encoder and decoder for the musical VQ representation. The adopted model has a hop length of 128, which corresponds to the top-level model from their original work Dhariwal et al. (2020). Jukebox employs 3 models: top-, middle-, and bottom-level, with both audio quality and required computation increasing from the first to the last model. As an example, in the supplemental HTML page, we provide music samples directly reconstructed from JukeBox using the top-level model we employ in our work, compared to the ground-truth audio. While they allow for high-quality audio reconstruction (from the bottom-level model, with a hop length of 8), it requires much more time and computation not only for training but also for the final inference, *e.g.*, *3 hours* to generate a *20-second* musical sequence. As the synthesized music from the top-level model includes some audible noise, we apply a noise reduction operation Sainburg et al. (2020). However, the overall audio quality is not a primary factor that we specifically address in this work on cross-modal conditioning and generation, as it largely depends on the specific music encoder and decoder that are employed. This explains why we report similar MOS scores in terms of the general audio quality.

### D.2 SYNTHESIZED IMAGES

We present more qualitative examples for text-to-image synthesis and class-conditioned image synthesis in Fig. 5, Fig. 6, and Fig. 7.

## E FURTHER DISCUSSION ON THE CDCD LOSS

In this section, we provide our further discussion on the proposed *CDCD* loss in terms of various aspects, including its relevance to the existing auxiliary losses, the impact of the *CDCD* strength, as well as additional experimental results.

### E.1 CDCD AS AUXILIARY LOSS

While the diffusion models are typically trained and optimized with the conventional variational lower bound loss $L_{vb}$ as we described in the main paper and Appendix B.2, there are several different types of auxiliary losses proposed to further regularize and improve the learning of diffusion models. Specifically, Dhariwal & Nichol (2021) introduces the idea of classifier based guidance for the diffusion denoising probabilistic models with continuous state space. Classifier-free guidance is proposed in Ho & Salimans (2022). In the area with discrete diffusion formulations Austin et al. (2021); Gu et al. (2022), an auxiliary loss that encourages the model to predict the noiseless token at the arbitrary step is adopted and proven to help with the synthesis quality.

Similar to the previous cases, we consider the proposed *CDCD* loss as a type of auxiliary losses, which seeks to provide additional guidance to better learn the conditional distribution $p(x|c)$. Specifically, the classifier-free guidance Ho & Salimans (2022) propose to randomly discard conditioning while learning a conditional diffusion generative model, which bears resemblance to our introduced downsampled contrastive steps in Appendix E.3.

### E.2 IMPACT OF CDCD STRENGTH

We further show the ablation studies on the parameter $\lambda$, which is the weight of our proposed *CDCD* loss that characterize the strength of this contrastive regularizer.

We conduct the dance-to-music generation experiments with different values of $\lambda$, and show the results in Tab. 9. As we observe from the table that the performance in terms of the beat scores are relatively robust for different $\lambda$ values ranging from $4e - 5$ to $5e - 5$. At the same time, we empirically observe that with a large value of $\lambda$, the model converges faster with less training epochs.

In case of the image synthesis task, we are rather cautious on the strength of the imposed contrastive regularizer. Intuitively, the proposed *CDCD* loss encourages the model to learn a slightly different distribution for negative samples, which could impose a trade-off between the one for the actual data given a specific conditioning. Therefore, while a larger value of $\lambda$ helps with the learning speed, we

Table 9: Ablation results in terms of $\lambda$ on the dance-to-music task.

| Value | Beats Coverage | Beats Hit |
|---|---|---|
| 5e-4 | 93.4 | 90.2 |
| 1e-5 | 93.3 | 91.0 |
| 5e-5 | 93.9 | 90.7 |

Table 10: Ablation results in terms of $\lambda$ on the dance-to-music task.

| Method | Beats Coverage | Beats Hit |
|---|---|---|
| Full ($T_c = 100$) | 93.9 | 90.7 |
| $T_c = 80$ | 93.4 | 90.5 |
| $T_c = 60$ | 92.7 | 90.4 |

empirically set the $\lambda$ to be $5e - 5$. Note that this value is adapted from the weight for other auxiliary losses in previous works Gu et al. (2022).

### E.3 DOWNSAMPLED CONTRASTIVE STEPS

While we show the performance of complete *step-wise* contrastive diffusion in the main paper, we discuss here an alternative way to implement the proposed method with less computational cost, by downsampling the contrastive steps in the diffusion process. Specifically, we randomly downsampled the steps with the proposed *CDCD* loss, which shares the similar spirit as in the class-free guidance Ho & Salimans (2022) to randomly drop out the conditioning. The experimental results are listed in Tab. 10, where there is little performance drop with downsampled contrastive steps.

