# OpenReview forum: "Discrete Contrastive Diffusion for Cross-Modal Music and Image Generation"
_ICLR.cc/2023/Conference — ICLR 2023 poster_

### Official Review · Reviewer_6GAW · 2022-10-24

**Confidence:** 3
**Correctness:** 3
**Technical Novelty And Significance:** 3
**Empirical Novelty And Significance:** 3
**Recommendation:** 6

**Clarity, Quality, Novelty And Reproducibility:**

###### Quality & Novelty

· The process of proof and experiment are sufficient.  Some people in the community will be very interested in this work.

###### Clarity:

· The author's writing is good and ideas were clarified well.

###### Reproducibility:

· The author provided implementation details  and already released their codes and pre-trained models.  I think this work is reproducible.

**Strength And Weaknesses:**

###### Strength：

· This work first combines diffusion training and contrastive learning, which provides a new perspective on the research of conditional DPMs.

· The proposed object function and contrastive diffusion mechanisms are shown promising quantitative and visual results.

· The frame work improves the convergence of diffusion models, reducing the number of required diffusion steps by more than 35% on two benchmarks.

· Proper INTRA- AND INTER-NEGATIVE SAMPLING methods are adopted and relevant experiments revealed their effectiveness.

###### Weakness：

I did't see any weaknesses

**Summary Of The Paper:**

This paper introduces a Conditional Discrete Contrastive Diffusion (CDCD) loss to enhance input-output connections by maximizing their mutual information. The author also designs two contrastive diffusion mechanisms to  incorporate  $L_{CDCD}$  into the denoising process.  Diverse multimodal conditional synthesis tasks have been evaluated and the proposed methods achieve higher or competitive general synthesis quality.


**Summary Of The Review:**


This paper introduces a simple yet effective method to improve the input-output connections via maximized mutual information. And both proposed DPMs are proven effective and achieve state-of-the-art scores in our evaluations. Compare to previous methods, the proposed methods show better performance. Overall, I think this paper is good.

---

### Official Review · Reviewer_3LEn · 2022-10-24

**Confidence:** 4
**Clarity, Quality, Novelty And Reproducibility:** Good.
**Correctness:** 3
**Technical Novelty And Significance:** 3
**Empirical Novelty And Significance:** 3
**Recommendation:** 8

**Strength And Weaknesses:**

Strengths:
To the best of my knowledge, this paper is the first to combine diffusion training and contrastive learning to design an effective generative model. The authors propose the CDCD loss and design two contrastive diffusion mechanisms to achieve this goal. The enhanced diffusion model is designed in a reasonable manner. The paper is well-written and well-organized. The proposed method is clearly described with all necessary analyses and mathematical details. Experiments have been conducted to demonstrate the effectiveness of the proposed method by showing higher synthesis quality and faster inference speed compared to other existing diffusion models.

Weaknesses:
By zooming in the synthesized images shown in Fig. 6 and Fig. 7, it seems to me that they are not as good as some SOTA results obtained by DALLE-V2, Imagen, etc. The authors need to explain the reasons and show/compare more qualitative results on the tasks of text-to-image synthesis and class-conditioned image synthesis in their supplemental materials.


**Summary Of The Paper:**

This paper proposes a so-called Discrete Contrastive Diffusion model for cross-modal music and image generation. The main idea is to explicitly enhance input-output connections by maximizing their mutual information instead of implicitly learning such relationships. Specifically, the authors try to combine diffusion training and contrastive learning via the conventional variational objectives. Experiments on three different tasks (i.e., dance-to-music generation, text-to-image synthesis, and class-conditioned image synthesis) verify the effectiveness of the proposed method by showing higher synthesis quality and faster inference speed compared to other existing diffusion models.



**Summary Of The Review:**

Due to enough novelty and academic contributions as analyzed above, I suggest accepting this paper after some minor revisions.

---

### Official Review · Reviewer_MzmQ · 2022-11-02

**Confidence:** 4
**Correctness:** 3
**Technical Novelty And Significance:** 3
**Empirical Novelty And Significance:** 3
**Recommendation:** 6

**Clarity, Quality, Novelty And Reproducibility:**

The idea that introducing the contrastive loss into the diffusion model is reasonable and novel which demonstrates the contribution of this paper. The proposed approach is clearly described in the paper.

**Strength And Weaknesses:**

Strength:
1. The idea of introducing contrastive loss to enhance the cross-modal relationship is reasonable. The proposed pipeline is well demonstrated in Figure2.
2. The proposed approach is clearly described, which makes the manuscript easy to follow.

Weaknesses:
1. The authors should provide more evidence to support the claim that incorporating prior into the variational lower bound can lead to the loss of the cross-modal correspondence.
2. Would the objective of enhancing cross-modal relationships contradict to increase the sample quality? How would the authors balance the variational loss and contrastive loss?
3. In the construction of inter-negative samples, the authors take all the images x’ other than x as negative samples. In this way, similar images may also be considered negative samples. How would the authors address this?
4. In the text-to-image generation task, the authors use the VQ-diffusion-S as the baseline. The results of the proposed approach slightly outperform the VQ-diffusion-S while falling behind the VQ-diffusion-B greatly. The authors should verify the effectiveness of the proposed approach on larger models.
5. In Table3, the performance of the proposed approach falls behind the DF-GAN.

**Summary Of The Paper:**

Existing diffusion-based cross-modal generation methods mainly establish the cross-modal relationships by incorporating the cross-modal prior model into the variational lower bound of the diffusion model. However, the authors claim that this method may lead to the loss of the cross-modal correspondence in the denoising process. To overcome this, the authors propose Conditional Discrete Contrastive Diffusion (CDCD) loss, which enhances the cross-modal relationships by constructing negative samples and introducing contrastive loss in the training.

**Summary Of The Review:**

The idea of this paper makes sense. However, the experimental results are not satisfactory. There are some issues with the proposed approach. Therefore, I believe that the paper is below the acceptance bar.

=======================================================================================================

Thank the authors for addressing my concerns. After reading the authors’ responses and peer reviews, I decide to change my score to 6. My main concern is the experiment results fall behind the VQ-D-B. The new experiment results show that the contrastive loss improves the FID of VQ-D-B from 19.7 to 18.44, which addresses my main concern.

---

### Official Review · Reviewer_uFmy · 2022-11-03

**Confidence:** 2
**Correctness:** 3
**Technical Novelty And Significance:** 3
**Empirical Novelty And Significance:** 3
**Recommendation:** 6

**Clarity, Quality, Novelty And Reproducibility:**

Clarity concerns:
- While I find the idea of using negative samples in a contrastive-based loss to improve a diffusion process interesting, the narrative is missing intuitive explanations of the “Step-Wise Parallel Diffusion” and “Sample-Wise Auxiliary Diffusion”. It is unclear from the paper the justification of the two different designs.
- Also, I think the proposed regularizer discussed in 3.1 requires more discussion. Is it fair to say that the regularizer enforces the diffusion process to also generate negative samples? From Fig. 2, I understand that the negative samples go through a denoising diffusion process too. Thus, is it fair to say that the regularizer thus makes the model learn two distributions: one distribution that generates the expected data given a condition, while the other one, is a distribution generating what it is not expected?

Reproducibility concerns:
- Given the current state of the paper, I find it hard to implement and reproduce the results. The architecture of the conditioning encoders is not discussed, and it is unclear if their parameters are optimized as part of the regularizer.
- Overall, the equations do not reveal what parameters are learned and are a bit convoluted. For example, in Sec. 3.1, the pdfs involved in the mutual information all seem to be parameterized by \theta. Are they really sharing the same parameters? Is f() the neural network to learn? If so, what is the architecture?


**Details Of Ethics Concerns:**

Given that the proposed method generates data, it may be used to generate fake data and used for harmful purposes.

**Strength And Weaknesses:**

Strengths:

I find the idea of using negative samples as part of the contrastive-loss-based regularizer to maximize the mutual information between the condition and latent interesting and novel. I think using these negative samples can indeed inform the model about what not to generate and constraint better the data generation.

Weaknesses:
1. While I find the idea novel, I think the method is quite elaborate and can imply more computational resources.
- First, including negative samples as part of the loss will increase computation making the computational cost even more expensive for a denoising diffusion process.
- Second, while the proposed stepwise diffusion allows parallelization, it still requires more resources that can increase the cost of an already expensive denoising diffusion process.

2. Insufficient experiments:
- The paper lacks an ablation study about the parameter $\lambda$ which controls the contribution to the total loss of the proposed regularizer. According to Section 4.1, $\lambda$ was set to 5e-5, which I find the value too low. It is unclear how to set this parameter from the experiments. More importantly, what the impact of increasing the value of $\lambda$ and thus enforcing the regularizer stronger on performance is not clear.
- The paper also misses an ablation study about the latent encoder. What is the effect of not even using one? Wouldn’t a latent encoder likely reduce the information (in the information theoretical sense) from the original input. Can the proposed methods work on raw signals, i.e., latents are the input signals directly.
- The experiments in paragraph “Results and Discussion” and Fig. 4 state that because the proposed method requires fewer steps to converge the proposed method is faster to converge. While the experiments show a reduction in steps, it is unclear about the cost of each step in terms of time in the proposed method. I think having a paper demonstrating that the proposed method indeed reduces the time of convergence is more important than the number of steps.
- The experiment in Table 3 is missing a more appropriate baseline: Stable Diffusion. I think using Stable Diffusion instead of DALLE makes more sense because Stable Diffusion also uses a latent representation while DALLE does not.

3. From the theoretical perspective, is there a proof showing that the proposed regularizer combined w/ the variational-bound-based loss still preserves the Langevin dynamics in some way? I think discussing the theoretical guarantees can be informative.

================= Post-Discussion =================

After engaging with the authors in the discussion, I still think the paper can benefit from reporting wall-clock time of the training phase, add more extensive ablation studies, and add Stable Diffusion as a baseline. For the most part, most of my concerns about clarity were addressed. Nevertheless, because I think there are missing experiments, I cannot champion the paper fully as I think the paper can benefit from another revision. I will slightly increase my rating to 6 - marginally above the acceptance threshold.

**Summary Of The Paper:**

The paper presents a regularizer that enforces that the condition embedding and the latent embedding of the input datum share the maximal mutual information between them. This regularizer then helps the diffusion process because the condition is rich with information about the latent embeddings, and thus helping the denoising process. The regularizer term is inspired by a contrastive loss in which the numerator indirectly measures the mutual information between the condition and latent embeddings, and the denominator includes information from conditions and embeddings that should not be related to each other (i.e., mimicking negative samples). The paper also discusses two ways of applying this regularizer: step-size parallel diffusion and sample-wise auxiliary diffusion. The former allows parallel evaluations of the denoising diffusion process while the latter does not. The paper presents experiments on dance-to-music generation, text-to-image synthesis, and class-conditioned image synthesis showing improvements over the included baselines.

**Summary Of The Review:**

Overall, I think the idea of using negative samples by means of a contrastive-based loss  is interesting. However, there are practical concerns that make me doubt the proposed method may be useful to speed up the diffusion processes. This is because while the proposed method indeed requires fewer steps, it is unclear if the step became more costly in terms of time or even computational resources. Second, the reproducibility given the state of the narrative falls short.

---

### Decision · Program_Chairs · 2023-01-20

**Decision:**

Accept: poster

**Justification For Why Not Higher Score:**

While it's difficult to conduct comparisons with much larger-scale models, it's unclear whether the proposed ideas would have the same impact at much larger scale.

**Justification For Why Not Lower Score:**

All the reviewers are positive about the paper, and two of them are very positive. R-6GAW said "I didn't see any weaknesses" yet only gave 6.

**Metareview: Summary, Strengths And Weaknesses:**

The paper proposed a novel conditional contrastive diffusion approach for better learning the statistical relationships between two modalities via the maximal mutual information. A contrastive regularizer is inspired by a contrastive loss where the numerator measures the mutual information between the condition and latent embeddings. The paper presented experimental results on diverse setups: dance-to-music generation, text-to-image synthesis and class-conditioned image synthesis, showing improvements over the baselines.

Overall, all the reviewers are positive about the paper, acknowledging the novelty of introducing contrastive learning in diffusion models. However, each reviewer shared concerns. R-uFmy mentioned the range of $\lambda$, whether negative samples would affect the training time, and Stable Diffusion should be benchmarked against. The author’s response has mostly addressed R-uFmy’s concerns, while R-uFmy still wants the authors to compare against Stable Diffusion because of the common nature of latent space, and has doubts whether the proposed method would require fewer steps. Consequently, R-uFmy raised the score to 6.

R-MzmQ raised the concerns of lack of evidence to support the claim that incorporating prior into the variational lower bound can lead to the loss of the cross-modal correspondence (an excellent outer-alignment question), how to balance the variational loss and contrastive loss (they seem to contradict), and other questions on the results and comparisons. Most of the issues were addressed in the rebuttal, and R-MzmQ also raised the score to 6.

R-3LEn raised the question of comparing the proposed CDCD with DALLE-2 and Imagen. However, it’s a bit unfair for a paper to compete against large-scale foundation models trained with a lot more resources.

Overall, the AC feels that the proposed idea has novelty and is backed up by solid experiments. The diverse experiments suggest that the proposed contrastive diffusion regularizers may have broader impact on the community.


**Note From Pc:**

if the above contains the word "oral" or "spotlight" please see: "oral" presentation means -> notable-top-5% and "spotlight" means -> notable-top-25%. As stated in our emails, we are disassociating presentation type from AC recommendations